



# Simultaneous Formation of Sulfate and Nitrate via Co-uptake of SO$_2$ and NO$_2$ by Aqueous NaCl Droplets: Combined Effect of Nitrate Photolysis and Chlorine Chemistry

5  Ruifeng Zhang[1,2], Chak Keung Chan[1,2,3]*

[1]School of Energy and Environment, City University of Hong Kong, Tat Chee Avenue, Kowloon 999077, Hong Kong, China
[2]Shenzhen Research Institute, City University of Hong Kong, Shenzhen 518057, China
10  [3]Low-Carbon and Climate Impact Research Centre, City University of Hong Kong, Tat Chee Avenue, Kowloon 999077, Hong Kong, China

*Correspondence to*: Chak K. Chan (chak.k.chan@cityu.edu.hk)

**Abstract.** SO$_2$ and NO$_2$ are the critical precursors in forming sulfate and nitrate in ambient particles. We studied the mechanism of sulfate and nitrate formation during the co-uptake of NO$_2$ and SO$_2$ into NaCl droplets at different RHs under irradiation and dark conditions. A significant formation of nitrate attributable to NO$_2$ hydrolysis was observed during the NO$_2$ uptake under all conditions, and its formation rate increases with decreasing RH. The averaged NO$_2$ uptake coefficient, 20  $\gamma_{NO2}$, from the unary uptake of NO$_2$ into NaCl droplets under dark are $1.6 \times 10^{-5}$, $1.9 \times 10^{-5}$, and $3.0 \times 10^{-5}$ at 80%, 70%, and 60% RH, respectively. Chloride photolysis and nitrate photolysis play a crucial role in sulfate formation during the co-uptake. Nitrate photolysis generates reactive species (e.g., OH radicals, NO$_2$, and N(III)) that directly react with S(IV) to produce sulfate. The generated OH radicals from nitrate photolysis can also react with chloride ions to form reactive chlorine species and then sulfate. To parameterize the role of nitrate photolysis and chloride photolysis in forming sulfate, the SO$_2$ 25  uptake coefficient, $\gamma_{SO2}$, as a function of nitrate photolysis rate, $P_{NO3-}$ ($= j_{NO3-} \times [NO_3^-]$), and chloride photolysis rate, $P_{Cl-}$ ($= j_{Cl-} \times [Cl^-]$), as $\gamma_{SO2} = 0.41 \times P_{NO3-} + 0.34 \times P_{Cl-}$ was derived. Our findings open up new perspectives on the formation of secondary aerosol from the combined effect of nitrate photolysis and chlorine chemistry.

## 1 Introduction

Sea salt aerosol (SSA) is one of the most abundant natural atmospheric particles in coastal environments (Chan and Yao 30  2008). Mainly composed of NaCl, it is generated from bursting bubbles during whitecap in the open ocean. In the atmosphere, fresh SSA can serve as reactive surfaces for the uptake of acidic gases such as SO$_2$, NO$_2$, and organic acids, followed by the release of HCl in the so-called "chloride depletion" reaction (Laskin et al., 2012; Yao et al., 2003). In the marine area or coastal region, the air pollutants, such as SO$_2$ and NO$_2$, emitted by substantial shipping emission has attracted





worldwide attention (Corbett et al., 2007; Zhang et al., 2019). These processes transform fresh SSA into aged SSA

containing NaNO$_3$, Na$_2$SO$_4$, and organic salts (Yao and Zhang 2012). In addition, chloride ions in the particles can also undergo reactions to produce a series of reactive chlorine species (e.g., Cl$^•$ and Cl$_2^{•-}$) that can significantly increase atmospheric oxidative capacity and formation of secondary pollutants (Gen et al., 2020; Wang and Ruiz 2017; Wang et al., 2020b; Young et al., 2014). In this study, we explore the co-uptake of SO$_2$ and NO$_2$ by NaCl droplets in producing sulfate and nitrate under irradiation and dark conditions.

Sulfate is one of the most abundant inorganic components in ambient particulate matter (Chan and Yao 2008). Although SO$_2$ emissions have been drastically reduced in China in the past decade (Zheng et al., 2018), the concentration of sulfate is still at a high level. On the other hand, NO$_2$ concentrations have not been reduced as significantly as SO$_2$ (Zheng et al., 2018). Due to the ubiquitous co-existence of SO$_2$ and NO$_2$ in the atmosphere, numerous experimental (Cheng et al., 2016; Ge et al., 2019; Li et al., 2018b; Liu and Abbatt 2021; Wang et al., 2016; Wang et al., 2020a) and theoretical (Tang

and Li 2021; Yang et al., 2019) studies have examined the role of NO$_2$ in the oxidation of SO$_2$ in/on aqueous particles and mineral dust surface. In aqueous particles, dissolved NO$_2$ reacts with S(IV) (= SO$_2$ + HSO$_3^-$ + SO$_3^{2-}$) to produce sulfate, S(VI):

S(IV) + 2 NO$_2$ (aq) + H$_2$O → S(VI) + 2 H$^+$ + 2 N(III) (aq)        (R1)

N(III) (NO$_2^-$/HONO) is produced as a by-product. In addition, gaseous NO$_2$ partitions into the aqueous phase and undergoes

hydrolysis (disproportionation) to produce nitrate and N(III) (R2). It has been reported that N(III) can also react with S(IV) to produce sulfate (Gen et al., 2019a; Wang et al., 2020a).

NO$_2$ (aq) + NO$_2$ (aq) + H$_2$O → N(III) (aq) + NO$_3^-$ (aq) + 2 H$^+$       (R2)

While sulfate formation via NO$_2$ oxidation under dark conditions has been widely reported, studies investigating the mechanisms of such processes under irradiation are scarce. Li et al. suggested that N(III) generated from the hydrolysis of

NO$_2$ can be photolyzed to produce OH radicals that can further react with S(IV) to form sulfate (Li et al., 2018b). Our earlier works (Gen et al., 2019a; Gen et al., 2019b; Zhang et al., 2020) reported effective SO$_2$ oxidation mediated by particulate nitrate photolysis, that is, oxidants (e.g., OH radicals, NO$_2$, and N(III)) generated from particulate nitrate photolysis react with S(IV) to yield sulfate.

The relative contribution of nitrate to PM pollution has increased in the past few years (Fu et al., 2020; Lin et al.,

2020; Xie et al., 2020). NO$_2$ is a key precursor of nitrate, and its role in forming nitrate has been well-documented. Prior studies reported the major formation pathways via gas-phase oxidation and subsequent gas-particle partitioning to produce nitrate (Alexander et al., 2009; Seinfeld and Pandis 2006). For instance, gas-phase oxidation of NO$_2$ by OH radicals to form nitric acid/nitrate. In addition, free nitrate radicals (NO$_3$), produced from the reaction of NO$_2$ with O$_3$, can also react with NO$_2$ to produce dinitrogen pentoxide (N$_2$O$_5$), which partitions into particles to form nitrate. Besides, the heterogeneous

uptake of NO$_2$ onto aqueous particles, mineral dust, and urban grime with subsequent hydrolysis (R2) to form nitrate has been reported (Dyson et al., 2021; Liu et al., 2019; Martins-Costa et al., 2020; Tan et al., 2016; Xu et al., 2019; Yu et al., 2021). The reactive uptake coefficient of NO$_2$, $\gamma_{NO2}$, is a crucial parameter controlling heterogeneous processes on the



aerosol surface. The contribution of $NO_2$ hydrolysis in forming nitrate from chemical transport models typically varies widely due to the significant uncertainties of $\gamma_{NO2}$ (Chan et al., 2021; Qiu et al., 2019b; Xie et al., 2022). On the other hand,

because much attention to R2 is principally due to the failure of the current atmospheric model to predict observed concentrations of HONO and OH radicals (Li et al., 2018a; Liu et al., 2019; Pandit et al., 2021; Xu et al., 2019), nitrate formation rate via $NO_2$ uptake and hydrolysis (R2) under various conditions, such as the presence of $SO_2$ and/or irradiation, has not been adequately addressed.

In this work, we studied sulfate and nitrate formation during the co-uptake of $SO_2$ and $NO_2$ by NaCl droplets under

dark and irradiation conditions (Figure 1). We reported an enhanced sulfate production rate during co-uptake of $SO_2$ and $NO_2$ under irradiation compared to dark. In addition, a significant amount of nitrate was formed under all conditions examined. A kinetic model was constructed to study the mechanisms of sulfate and nitrate formation. We found that the interaction between nitrate photolysis and chlorine chemistry plays an important role in sulfate formation. The factors affecting the sulfate and nitrate formation rates were also discussed.

## 2 Method and Materials

### 2.1 Materials

Aqueous stock solutions of sodium chloride (NaCl; 99.8%, Uni-Chem), sodium nitrate ($NaNO_3$; >99%, Acros Organics), sodium sulfate ($Na_2SO_4$; >99%, Acros Organics), and ammonium sulfate (($NH_4$)$_2SO_4$; >99%, VWR Chemicals BDH) were prepared by dissolving corresponding salts into ultrapure water. The stock solution was atomized to generate droplets by a

droplet generator (MicroFab Technologies, S.N. JD5-1008), and individual droplets of a diameter of ($57\pm2$) μm were collected on a transparent hydrophobic substrate (model 5793, YSI, Inc.) in a flow cell. The deposited droplets were equilibrated for ~30 min at a given RH before each reactive uptake experiment. Although the droplets used in this study are larger than ambient particles, we analyzed the kinetic data using uptake coefficients, which have taken size effects into consideration. The irradiation experiments were initiated using a xenon lamp (model 6258, Ozone free xenon lamp, 300 W,

Newport) equipped with a long-pass filter (20CGA 305 nm cut-on filter; Newport) to eliminate light below 300 nm. The averaged initial photon fluxes at 280 nm to 420 nm received by droplets were ~4.0 × 10$^{16}$ photons cm$^{-2}$ s$^{-1}$ using 2-nitrobenzaldehyde as a chemical actinometer (Gen et al., 2021; Gen et al., 2020).

### 2.2 Reactive Uptake Experiments and In-Situ Raman Characterization.

The reactive uptake of $SO_2$ and $NO_2$ into NaCl droplets experiments were performed using a Raman microscope/flow cell

setup (Figure S1 in the Supporting Information), described in detail in our previous studies (Gen et al., 2021; Gen et al., 2019a; Gen et al., 2019b; Gen et al., 2020; Zhang et al., 2021; Zhang et al., 2020; Zhang et al., 2022). Here, we give a brief description. The reactive uptake experiments were performed with $SO_2$ and $NO_2$ concentrations of ~6.5 ppm and ~10 ppm,





respectively, under controlled relative humidity (RH) and light/dark conditions. The RH in the flow cell was controlled by adjusting wet and dry synthetic air flow rates. We studied the uptake process and its subsequent reactions by in-situ

characterizing the variation of particle composition by Raman spectroscopy (EnSpectr R532, EnSpectr). The Raman shift at ~980, ~1050, and ~3400 cm$^{-1}$ are assigned to $\nu(SO_4^{2-})$, $\nu(NO_3^-)$, and $\nu(OH)_{water}$, respectively. The change in concentration of $[SO_4^{2-}]$ and $[NO_3^-]$ in droplets was quantified by using established calibration curves (Figure S2). The reacted droplets were dissolved in ~1 mL ultrapure water, followed by ion chromatograph analysis (IC analysis) with an IonPac AS15 analytical column, an AG15 guard column, and a conductivity detector.

**2.3 Kinetic Simulations.**

A kinetic model was constructed in a chemical kinetics simulation package (FACSIMILE) to better understand reaction mechanisms. The reactions listed in Table S1 were used in the kinetic model. The model simulation results are shown in Figure S3, in general, the agreement between the observed and predicted sulfate and nitrate concentrations was good. The initial chloride concentrations at different RHs were estimated based on the extended-aerosol inorganic model (E-AIM)

(Clegg et al., 1998). Note that chloride loss may occur in the form of HCl at equilibrium; hence, the corrected chloride concentrations based on IC analysis were used as input in the model (Table S2). We performed experiments with unary uptake of $NO_2$ into NaCl droplets under dark to monitor the chloride depletion due to the increased acidity. As shown in Figure S4, the molar ratio of $Cl^-$ to $Na^+$, $n(Cl^-)/n(Na^+)$, decreased during the reaction. We simulate the evaporation of HCl based on the IC results to obtain the depletion rate of $Cl^-$ ($Cl^- \rightarrow HCl\ (g)$), which was also incorporated into the kinetic model

in all conditions. The nitrate photolysis rate constant, $j_{NO3-}$, and the chloride photolysis rate constant of $Cl^- + h\nu \rightarrow Cl^\bullet$, $j_{Cl-}$, were used as the fitting parameters to reproduce the observed changes in sulfate and nitrate concentrations. The fitted $j_{NO3-}$ and $j_{Cl-}$ are on the order of $10^{-6}$ s$^{-1}$ and $10^{-7}$ s$^{-1}$ (Table S3), respectively, which fall in the range reported in the literature (Gen et al., 2019a; Kalmár et al., 2014; Ye et al., 2017; Zhang et al., 2020). In addition, the effective Henry's law constant of $NO_2$ and $SO_2$, denoted as $H_{NO2}^*$ and $H_{SO2}^*$, respectively, may vary due to the reactions in the droplets. Equations of $H_{NO2}^*$ and $H_{SO2}^*$

as a function of time were established based on the observed time profiles of nitrate and sulfate (Text S1), which were incorporated in the model simulation.

**3 Results and discussions**

**3.1 Mechanisms of Sulfate Formation Under Irradiation.**

As displayed in Figure 2a, no observable sulfate formation was found during the co-uptake of $NO_2$ and $SO_2$ into aqueous

NaCl droplets at 80% RH under dark, indicating that direct aqueous oxidation of S(IV) by $NO_2$ (R1) might not make a significant contribution to sulfate in the present study. The same was observed at all RH values (Figure 3a). This contrasts with some prior studies on significant sulfate formation from the oxidation of S(IV) by $NO_2$ during the polluted period (Liu



and Abbatt 2021; Wang et al., 2016; Wang et al., 2020a). The negligible sulfate formation in our study may be due to the rapid drop in pH (Figure S5). Specifically, $NO_2$ uptake and subsequent hydrolysis release $H^+$ (R2), increasing particle acidity, which limits the dissolution of $SO_2$ (Seinfeld and Pandis 2006). Figure S5 shows pH decreases to about 2 within 240 min due to the $NO_2$ hydrolysis.

The co-uptake of $SO_2$ and $NO_2$ under irradiation gave an enhanced sulfate formation rate (~$2.7 \times 10^{-6}$ M•s$^{-1}$; Figure 2a) compared to that under dark at 80% RH, which suggests there are additional photochemical pathways to sulfate production. Several pathways of sulfate formation might explain these observations. First, O($^3$P) atom generated from $NO_2$ (g) photolysis ($\lambda \leq 420$ nm) can react with $O_2$ to form $O_3$ (g) in air ($O_2 + O(^3P) \rightarrow O_3$) (Gardner et al., 1987; Trebs et al., 2009). Dissolved $O_3$ in aqueous particles can react with S(IV) to form sulfate (Seinfeld and Pandis 2006). However, our open system experiments with a continuous airflow (~ 0.5 L/min) would have removed the formed $O_3$ efficiently. Second, reactive chlorine species (e.g., Cl$^•$ and Cl$_2$$^{•-}$) can be formed by photoinduced electron transfer from chloride ions, denoted as "chloride photolysis ("CP" in short)" hereafter (Grossweiner and Matheson 1957; Kalmár et al., 2014; Zhang and Parker 2018). These chlorine species can react with S(IV) to produce sulfate (Table S1) (Zhang et al., 2020). As shown in Figure 2a, the formation of sulfate (~$1.3 \times 10^{-6}$ M•s$^{-1}$) was observed during the unary uptake of $SO_2$ into NaCl droplets under irradiation. No sulfate peak was detected in NaCl droplets without irradiation in the presence of $SO_2$ alone (Figure 2a). In addition, lower light intensity yields a slower sulfate formation from the unary uptake of $SO_2$ (Figure S6 and Table S4), further confirming that the chloride photolysis process can drive the sulfate formation. It should be noted that sole chloride photolysis-driven sulfate production cannot explain the observed higher sulfate production rate during the co-uptake (Figure 2a). Third, N(III) (HONO/$NO_2^-$), produced from the reaction of $NO_2$ with S(IV) (R1) (Ge et al., 2019; Liu and Abbatt 2021; Tang and Li 2021; Wang et al., 2020a) and $NO_2$ hydrolysis (R2) (Yabushita et al., 2009), can directly react with $HSO_3^-$ to produce sulfate (Wang et al., 2020a), or indirectly undergo photolysis to produce OH radicals that further promote sulfate formation (Seinfeld and Pandis 2006). Interestingly, no $NO_2^-$ Raman peak was observed in all experiments (Figure S7). IC analysis also confirmed negligible $NO_2^-$ in the reacted droplets (Figure S8). We postulated that HONO from the protonation of $NO_2^-$ partitions into the gas phase at low pH (Figure S5) and can be removed rapidly because of the open flow cell system. Figure S5 shows that the droplets pH decreases below the pKa of $NO_2^-$/HONO (~3.2) (Arakaki et al., 1999) within 240 min due to the $NO_2$ hydrolysis. Li et al. found that more than 95% of $NO_2$ was hydrolyzed to form HONO and nitrate at the surface of aqueous sodium sulfite microjets (Li et al., 2018b). The effective mass transfer of HONO between particle and gas makes the accumulation of nitrite inside of particle negligible. Thus, HONO/$NO_2^-$ is not expected to have significantly contributed to the sulfate production.

On the contrary, significant nitrate production was found during the co-uptake under both light and dark conditions (Figure 4a). Our previous works (Gen et al., 2019a; Gen et al., 2019b; Zhang et al., 2020) reported that particulate nitrate photolysis ("NP" in short) could effectively promote the oxidation of $SO_2$ to form sulfate. We attributed the further enhanced sulfate production during the co-uptake compared to the unary uptake of $SO_2$ to nitrate photolysis (Figure 2a). First, the reactive species (e.g., OH, $NO_2$, and N(III)) produced from nitrate photolysis (denoted as "NP+ONN" reactions) (i.e., SR1 in





Table S1) can directly react with S(IV) to form sulfate, denoted as $[SO_4^{2-}]_{NP+ONN}$ pathway (Gen et al., 2019a; Gen et al., 2019b). Second, the OH radicals generated from nitrate photolysis can react with $Cl^-$ to yield the reactive chlorine species (e.g., $Cl^\bullet$ and $Cl_2^{\bullet-}$; denoted as "NP+Cl" reactions) (i.e., SR41, SR42, and SR43 in Table S1), that could further enhance

sulfate production, denoted as $[SO_4^{2-}]_{NP+Cl}$ pathway. Note that $[SO_4^{2-}]_{NP+Cl}$ pathway does not require chloride photolysis. In addition, we previously reported enhanced sulfate production from the halide-induced enhancement of nitrate photolysis by attracting more nitrate towards the interface, where an incomplete solvent cage leads to an increase in the quantum yield of oxidants from nitrate photolysis (Zhang et al., 2020). Such an effect might be embedded in the two processes mentioned above, especially at $[Cl^-]/[NO_3^-] \leq 0.2$, as proposed in our earlier work (Zhang et al., 2020). However, the model predicted

$[Cl^-]/[NO_3^-]$ at all studied RHs were higher than 0.2 in this study, hence the halide-induced enhancement of nitrate photolysis is minor within the simulation time scale in the kinetic model. The contribution of different pathways to sulfate production will be discussed later. The role of nitrate photolysis in sulfate formation, $[SO_4^{2-}]_{NP}$, hereafter refers to the sum of $[SO_4^{2-}]_{NP+ONN}$ and $[SO_4^{2-}]_{NP+Cl}$ pathways, except when stated otherwise.

We used the kinetic model to simulate the time series of sulfate concentration during the co-uptake of $NO_2$ and $SO_2$

under irradiation at 80% RH to understand the sulfate formation mechanisms. The contributions of chloride photolysis, $[SO_4^{2-}]_{CP}$, and nitrate photolysis (i.e., $[SO_4^{2-}]_{NP} = [SO_4^{2-}]_{NP+ONN} + [SO_4^{2-}]_{NP+Cl}$) to sulfate production were evaluated in the kinetic model. Figure 2b shows $[SO_4^{2-}]_{CP}$ pathway dominates over the $[SO_4^{2-}]_{NP}$ before ~1200 min, due to the high concentration of chloride. The contribution of $[SO_4^{2-}]_{CP}$ pathway to total sulfate production continuously decreases. In contrast, the fraction of sulfate concentration generated from $[SO_4^{2-}]_{NP}$ pathway shows an increasing trend, yielding ~58% of

total sulfate after 1440 min (Figure 2c). We further investigate the role of the $[SO_4^{2-}]_{NP+ONN}$ and $[SO_4^{2-}]_{NP+Cl}$ pathways in the formation of sulfate. As shown in Figure 2b, the $[SO_4^{2-}]_{NP+ONN}$ and $[SO_4^{2-}]_{NP+Cl}$ pathways contribute to ~18% and ~40% of total sulfate production, respectively, after 1440 min. For the $[SO_4^{2-}]_{NP+ONN}$ pathway, the N(III) pathway dominates over the $NO_2$ pathway and OH pathway in forming sulfate (Figure S9), which is consistent with our previous works (Gen et al., 2019a; Gen et al., 2019b). The role of $[SO_4^{2-}]_{NP+ONN}$ relative to $[SO_4^{2-}]_{NP+Cl}$ pathway becomes important at a later stage. As

shown in Figure 2c, $[SO_4^{2-}]_{NP+ONN} / [SO_4^{2-}]_{NP}$ and $[SO_4^{2-}]_{NP+Cl} / [SO_4^{2-}]_{NP}$ show increasing and decreasing trends as the reaction proceeds, respectively, which is likely due to the significant chloride depletion near the end (Figure S4). For the $[SO_4^{2-}]_{NP+Cl}$ pathway, we postulated that OH radicals could promote the formation of reactive chlorine species that can further react with S(IV) to form $SO_3^-$ and then sulfate. Specifically, in the presence of nitrate photolysis and chloride, the OH radicals from nitrate photolysis can react with $Cl^-$ to yield $ClOH^-$ that combines with $H^+$ to generate $Cl^\bullet$ radicals. The

produced $Cl^\bullet$ further reacts with $Cl^-$ to yield $Cl_2^{\bullet-}$, and both can react with $HSO_3^-$ to yield $SO_3^-$, which undergoes chain reactions to produce sulfate (Table S1). Typically, $SO_3^-$ radicals are mainly produced from the reaction of $OH + HSO_3^-$ or $Cl^\bullet/Cl_2^{\bullet-} + HSO_3^-$ in this study. Figure S10 displays that $SO_3^-$ formed from $OH + HSO_3^-$ is significantly lower than that from $Cl^\bullet/Cl_2^{\bullet-} + HSO_3^-$, further confirming that chlorine chemistry plays a vital role in sulfate formation. Note that $Cl^\bullet/Cl_2^{\bullet-}$ can be generated from chloride photolysis and "NP+Cl" process. Figure 2d shows that chloride photolysis only contributes to the

significant formation of $SO_3^-$ radical at the initial stage, and the decreasing trend of $SO_3^-$ indicates that the consumption rate





of $SO_3^-$ is higher than the formation rate without the assistance of nitrate photolysis. In contrast, the contribution of nitrate photolysis to $SO_3^-$ concentration increases and dominates over chloride photolysis after ~750 min (Figure 2d). These results at 80% RH highlighted that interaction between nitrate photolysis and chlorine chemistry is crucial in enhancing sulfate formation.


### 3.2 The Effect of RH and Presence of $NO_2$ in Forming Sulfate.

As discussed earlier, the direct reaction of $NO_2$ with S(IV) under dark is not effective in producing sulfate at 80% RH in this study. Also, no sulfate was observed during the unary uptake of $SO_2$ into NaCl droplets under dark (Figure 2a). The same was found at 60% and 70% RH (Figure 3a). Therefore, we focused on the RH dependence of sulfate production rates in the

presence of irradiation.

As shown in Figure 3a and Figure S11, the sulfate production rate increases with decreasing RH, which is attributed to the increased concentrations of oxidants at low RH, irrespective of the presence of $NO_2$ or not. For unary $SO_2$ uptake experiments, the sulfate production rate increases from $1.3 \times 10^{-6}$ to $3.3 \times 10^{-6}$ M•s$^{-1}$ when RH decreases from 80% to 60% RH (Table S4). During the co-uptake course of $NO_2$ and $SO_2$, sulfate concentrations tend to follow a sigmoidal trend with a

slow initial increase followed by a rapid one before slowing down, especially at low RH (Figure 3a). Table S4 shows the averaged sulfate production rates of the initial and the fast-growing stages. The sulfate production rate at the initial stage is comparable to that observed during the unary uptake of $SO_2$ by NaCl particles under irradiation at all RHs (Figure S12 and Table S4), indicating chloride photolysis plays a major role in forming sulfate, where nitrate concentration is less than 1M (Figure 4a). Such low nitrate concentration cannot significantly affect sulfate production. Enhanced sulfate production was

observed at the second stage when nitrate concentration reached up to a few Ms. This enhancement resulted from the nitrate photolysis and its interaction with chlorine chemistry, as discussed earlier. The sulfate production rate at the second stage increases from $2.7 \times 10^{-6}$ to $8.6 \times 10^{-6}$ M•s$^{-1}$ as RH decreases from 80% to 60% RH (Table S4). The slower sulfate production rate near the end is likely due to significant chloride consumption resulting from photolysis and evaporation. Figure 2b and Figure S13 show the predicted sulfate concentration from $[SO_4^{2-}]_{CP}$, $[SO_4^{2-}]_{NP}$, $[SO_4^{2-}]_{NP+Cl}$, and $[SO_4^{2-}]_{NP+ONN}$

pathways at 80%, 70%, and 60% RH. The model simulation time are ~660 min, ~1200 min, and ~1440 min for 60%, 70%, and 80% RH, respectively (Figure 2b and Figure S13), after which nitrate concentrations have increased beyond the range of the calibration shown in Figure S2. The $[SO_4^{2-}]_{CP}$ contributes a large fraction (> 95%) of total sulfate production before 250 min at all RHs (Figure 2c and Figure S13), consistent with observations from experiments (Figure S12), further confirming that chloride photolysis plays a dominant role in sulfate formation in the early stage. The role of $[SO_4^{2-}]_{NP}$ appears to be more

important at a later stage (Figure 2c and Figure S13), especially at low RH, because of the higher nitrate concentration. As shown in Figure 3b, $[SO_4^{2-}]_{NP}$ / $[SO_4^{2-}]_{CP}$ increases faster at low RH. In addition, $[SO_4^{2-}]_{NP + Cl}$ to $[SO_4^{2-}]_{NP + ONN}$ ratio of significantly larger than unity was found in all studied RHs, suggesting the combined effect of nitrate photolysis and chlorine chemistry exerts a crucial impact in enhanced sulfate formation (Figure 3b).



Furthermore, the reactive uptake coefficients of $SO_2$, $\gamma_{SO2}$, were compared for unary and co-uptake to investigate

the impact of $NO_2$ in sulfate formation. We focused on the irradiation results because there was no evident sulfate production

under dark irrespective of $NO_2$ presence or not (Figure 2a). It should be noted the $\gamma_{SO2}$ of co-uptake shown in Figure 3c is

derived from the fast-growing (second) stage of sulfate formation (Figure S12), the regime where the combined effect of

nitrate photolysis and chlorine chemistry plays a crucial role. The $\gamma_{SO2}$ increases with decreasing RH, with larger impacts in

the presence of $NO_2$ (Figure 3c and Table S5). In the absence of $NO_2$, enhanced $\gamma_{SO2}$ is mainly resulted from the increased

reactive chlorine species produced from chloride photolysis at high concentrations of chloride at low RH. The presence of

$NO_2$ significantly increased $\gamma_{SO2}$ (Figure 3c) further due to nitrate photolysis. The $\gamma_{SO2}$ of co-uptake increased by a factor of

~1.7, ~2.5, and ~2.9 at 80%, 70%, and 60% RH, respectively, compared to that of unary uptake. These results imply that the

effect of $NO_2$ in forming sulfate becomes more important at low RH. To highlight the importance of the effect of nitrate

photolysis and chloride photolysis in forming sulfate due to the presence of $NO_2$, we developed an equation of $\gamma_{SO2}$ as a

function of nitrate photolysis rate, $P_{NO3-}$ ($= j_{NO3-} \times [NO_3^-]$), and chloride photolysis rate, $P_{Cl-}$ ($= j_{Cl-} \times [Cl^-]$), as $\gamma_{SO2} = 0.41 \times$

$P_{NO3-} + 0.34 \times P_{Cl-}$, based on the experimental results from the co-uptake of $NO_2$ and $SO_2$ under irradiation at three RHs. The

derived expression matched with the experimental datapoints well (Figure S14). It should be noted that the derived

expression is constrained to the conditions in the presence of nitrate and chloride, especially at $[NO_3^-]/[Cl^-] < 3$.

**3.3 Mechanisms of Nitrate Formation during the Uptake.**

As discussed, there was significant nitrate formation under both dark and irradiation.  Under dark, it is mainly produced from

$NO_2$ hydrolysis regardless of the presence of $SO_2$ or not, hereafter referred to as the "$NO_2$ hydrolysis pathway". Previous

studies reported the reaction of $NO_2$ with $Cl^-$ ($2\ NO_2\ (aq) + Cl^-\ (aq) \rightarrow NO_3^-\ (aq) + ClNO\ (g)$), which can also yield nitrate

(Karlsson and Ljungström 1998; Weis and Ewing 1999). To investigate whether this process is also responsible for the

significant formation of nitrate under dark, we performed a control experiment with unary uptake of $NO_2$ into $(NH_4)_2SO_4$

droplets at 80% RH. As shown in Figure S15, prompt nitrate formation was also observed in $(NH_4)_2SO_4$ droplets at a rate

just slightly lower than NaCl droplets. This result indicates that the role of the reaction of $NO_2$ with chloride in forming

nitrate is minor, but $NO_2$ hydrolysis plays a dominant role in the formation of nitrate under dark. During the co-uptake, the

available S(IV) without being oxidized under dark can further promote $NO_2$ uptake and nitrate formation (Figure 5b), as will

be discussed later. Under irradiation, $NO_2$ from reactive uptake can also react with OH radicals or $NO_2$ from nitrate

photolysis can form nitrate, denoted as "$NO_2$ + OH pathway" and "$NO_2$ + $NO_2$ pathway", respectively. Besides, $NO_3$

radicals can undergo radical chain reactions to produce nitrate (Table S1), referred to as the "$NO_3$ pathway".

Figure 4a shows the nitrate concentration trends of co-/unary uptake of $NO_2$ under irradiation/dark conditions at

80% RH. A significant amount of nitrate was observed via the reactive uptake of $NO_2$ into NaCl droplets. The presence of

$SO_2$ has little effect on $NO_2$ uptake at 80% RH. Under dark condition, an exponential increase in nitrate production was



observed, particularly at low RH (Figure 4a and Figure 5a). The pH in these dark experiments decreased during the uptake (Figure S5). However, it is known that increased acidity (i.e., decreased pH) due to hydrolysis is not conducive to $NO_2$ uptake (Seinfeld and Pandis 2006). We speculated that chloride depletion plays a mechanistic role in the exponential increase in nitrate concentration. $NO_2$ uptake converts NaCl to $NaNO_3$, which lowers the hygroscopicity of the droplet

(Clegg et al., 1997). resulting in the droplet shrinkage of ~10% after 1440 min uptake (Figure S16). We used E-AIM model to estimate chloride and nitrate concentrations at different molar ratios of chloride to nitrate, $[Cl^-]/[NO_3^-]$, at a given RH. As shown in Figure S17, nitrate concentration increased exponentially when $[Cl^-]/[NO_3^-]$ decreased below 3 due to the lower hygroscopicity of $NaNO_3$ compared to NaCl, i.e., decreasing liquid water content of the droplet, which may partially explain the observed exponential increase in nitrate concentration in Figure 4a and Figure 5a. The overall trend is that the nitrate

production rate is higher under dark than under irradiation, which may be partly due to nitrate photolysis at 80% RH.

Furthermore, we examined the time series of nitrate concentration during the co-uptake at 80% RH under irradiation to elucidate the nitrate formation mechanism using the kinetic model. Figure 4b and 4c show the contributions of different pathways to nitrate during the co-uptake of $NO_2$ and $SO_2$ under irradiation. The $NO_2$ hydrolysis pathway yields ~2.1 M nitrate, but the $NO_2 + NO_2$ pathway, $NO_2 + OH$ pathway, and $NO_3$ pathway altogether contribute to only ~0.02 M nitrate

after 1440 min. Hence, $NO_2$ hydrolysis is dominant in nitrate formation. The presence of chloride and S(IV) can consume the oxidants (e.g., OH radicals and chlorine species), potentially making those three pathways ineffective in forming nitrate (Table S1). Among these three pathways, the $NO_2 + NO_2$ pathway (left axis, Figure 4c) contributes most to nitrate, and the other two pathways are negligible (right axis). Note that ~0.1 M nitrate was consumed due to nitrate photolysis (i.e., SR1 in Table S1) after 1440 min. This minor consumption of nitrate due to photolysis is related to the low nitrate photolysis rate

constant (~$1.4 \times 10^{-6}$ s$^{-1}$ at 80% RH), based on fitting the experimental results, which is about one order of magnitude lower than that fitted in our previous works (Gen et al., 2019b; Zhang et al., 2021; Zhang et al., 2020). The photolysis rate constant is typically proportional to the light intensity. Also, it is known that nitrate has a maximum absorption band at ~302 nm (Gen et al., 2022). The light intensity at 300 nm of the Xenon lamp used in the current study is around 3 orders of magnitude lower than that of the single-line 300-nm lamp used in our earlier works.(Gen et al., 2020)

In the unary uptake of $NO_2$ into NaCl droplets under irradiation, $NO_2$ hydrolysis still dominates over those three pathways initiated by nitrate photolysis (Figure S18). However, in addition to the $NO_2 + NO_2$ pathway, the $NO_3$ pathway also makes a non-negligible contribution to nitrate formation during the unary uptake of $NO_2$ compared to that during the co-uptake. The reaction of $Cl_2^{\bullet-}$ with $NO_3$ (SR50 in Table S1) is a dominant process of $NO_3$ pathway. However, in the co-uptake experiments, the consumption of $Cl_2^{\bullet-}$ by S(IV) (SR59 in Table S1) makes nitrate production by the reaction of chlorine

species with $NO_3$ radicals ineffective (Figure S19).



### 3.4 The Effect of RH and Presence of SO₂ in Forming Nitrate.

We further investigate the effect of RH on the nitrate production rate during the uptake. In all experiments, including unary/co- uptake under dark/irradiation, the nitrate production rate increased with decreasing RH (Figure 5 and Figure S20),
which is attributed to the increased NaCl concentration at low RH. A prior study reported an enhanced uptake coefficient of NO₂ at increasing concentration of NaCl solutions (Yabushita et al., 2009). Figure 5a shows the nitrate formation during the co-uptake under both dark and irradiation at different RHs. The nitrate concentration increased slowly initially, with a faster increase at a later stage at all RHs, attributable to chloride depletion, as discussed earlier. Under dark, NO₂ hydrolysis and the reaction of NO₂ with S(IV) are the only two reactions to form nitrate in droplets, with the former being dominant. Under
irradiation, similar to the 80% RH experiments, the NO₂ hydrolysis pathway dominated the contribution to nitrate at 60% and 70% RH (Figure S21). The nitrate concentration under irradiation during the co-uptake was lower than those under dark at all RHs. However, the difference between dark and irradiation became smaller with decreasing RH. We speculate that the faster nitrate production at low RH can compensate for the consumption of nitrate due to photolysis. Such phenomena were also observed during the unary uptake under dark and irradiation (Figure S20a).

We also investigated the effect of SO₂ on nitrate production under dark and irradiation. Figure 5b shows that the presence of SO₂ promotes nitrate formation under dark, especially at low RH. We proposed that such enhancement might be closely associated with the reaction between NO₂ and HSO₃⁻. The increase in HSO₃⁻ peak intensity during unary uptake of SO₂ into NaCl under dark indicates the formation of HSO₃⁻ from SO₂ dissolution is feasible in our system (Figure S22). The formed HSO₃⁻ from reactive uptake of SO₂ into particles prefers to stay at the surface of an aerosol particle (Yang et al.,
2019), and in turn, it may pull more NO₂ into droplets during the co-uptake due to the interaction between NO₂ and HSO₃⁻ (Tang and Li 2021). The higher formation rate of HSO₃⁻ at low RH during unary SO₂ uptake into NaCl may explain the more considerable difference in nitrate concentration between co- and unary uptake at low RH than at 80% RH (Figure 5b). In contrast, nitrate concentrations between co- and unary uptake under irradiation are comparable (Figure S20b). We proposed that HSO₃⁻ may undergo photooxidation rapidly to convert into sulfate, hence its role in pulling more NO₂ into droplets
becomes limited under irradiation during the co-uptake. These findings open new perspectives on the enhanced formation of nitrate with the involvement of SO₂, especially at low RH.

### 4 Atmospheric Implication

Built on the well-known process of SO₂ oxidation by NO₂ in forming sulfate under dark, we examined sulfate formation during the co-uptake of SO₂ and NO₂ in NaCl droplets under irradiation in this work. In our earlier works, we examined the
effect of nitrate photolysis in forming sulfate and glyoxal-derived SOA (Gen et al., 2019a; Gen et al., 2019b; Zhang et al., 2021; Zhang et al., 2020; Zhang et al., 2022). The present study indicated a previously unrecognized role of the combined effect of nitrate photolysis and chlorine chemistry in enhanced sulfate formation, which is expected to have important implications on atmospheric chemistry. In brief, the OH radicals formed from nitrate photolysis can react with chloride ions





to promote the formation of chlorine species (e.g., Cl$^\bullet$ and Cl$_2^{\bullet-}$) and further enhance sulfate formation. Such effect enhances

the $\gamma_{SO2}$ by factors of ~1.7, ~2.5, and ~2.9 at 80%, 70%, and 60% RH, respectively, compared to those from the chloride photolysis (i.e., NaCl + SO$_2$ + light conditions). In our earlier work, we performed unary uptake of SO$_2$ into premixed NH$_4$Cl + NH$_4$NO$_3$ droplets under 300-nm irradiation at different ratios of Cl$^-$ to NO$_3^-$, [Cl$^-$]/[NO$_3^-$], in air and N$_2$ environment (Zhang et al., 2020). Note that the presence of O$_2$ (air environment) is essential for sulfate formation from reactions of reactive chlorine species with HSO$_3^-$ (Figure 1). The normalized sulfate production rate continuously increases as [Cl$^-$

]/[NO$_3^-$] increases in air, but no further enhancement of sulfate production rate was observed in N$_2$ when [Cl$^-$]/[NO$_3^-$]>0.2 (Zhang et al., 2020). Such results also support the significance of the combined effect of nitrate photolysis and chlorine chemistry in promoting sulfate formation. These results highlighted that the coexistence of nitrate photolysis and chlorine chemistry can increase the atmospheric oxidative capacity and enhance the formation of sulfate. Further investigations into the combined effect of nitrate photolysis and chlorine chemistry in forming SOA are proposed.

335         Prior studies have pointed out that chlorine atom (Cl$^\bullet$) can play an essential role in increasing atmospheric capacity and the formation of secondary pollutants and ozone (Li et al., 2021; Liao et al., 2014; Qiu et al., 2019a). Typically, Cl$^\bullet$ can form from the photo-dissociation and oxidation of inorganic chlorine species (e.g., Cl$_2$ and ClNO$_2$) and chlorinated organic species (e.g., CHCl$_3$ and CH$_2$Cl$_2$) (Priestley et al., 2018; Wang et al., 2020b). Cl$^\bullet$ is much more reactive than OH radical (Hossaini et al., 2016; Su et al., 2022), although it typically has lower concentrations and is in limited regions of the

atmosphere compared to the OH radicals (Saiz-Lopez and von Glasow 2012; Wang and Hildebrandt Ruiz 2018). Laboratory studies have shown the importance of Cl$^\bullet$ on the oxidation of VOC precursors (e.g., common monoterpenes, isoprene, toluene, and alkanes) in forming SOA (Dhulipala et al., 2019; Masoud and Ruiz 2021; Riva et al., 2015; Wang and Ruiz 2017). The rate constant of Cl$^\bullet$ with most atmospherically relevant VOCs is up to 2 orders of magnitude higher than the rate constant of OH-initiated oxidation (Masoud and Ruiz 2021). In addition to nitrate and sulfate formation, the current study

provides new insight into the enhanced formation of chlorine species due to nitrate photolysis. Recently, Peng et al. proposed that nitrate photolysis at high aerosol acidity is an important pathway for activating inert chloride to produce photolabile Cl$_2$ during daytime in the polluted period, which is a highly reactive species and can strongly affect the abundance of climate and air quality-relevant traces gases (Peng et al., 2022). Same as the reactions suggested in this study, they also suggested that OH radicals produced from nitrate photolysis can further oxidize chloride ions to enhance the formation of Cl$_2$ (Peng et

al., 2022). Our kinetic model also indicates that Cl$^\bullet$ generated from chloride photolysis could also undergo self-reaction to form Cl$_2$. However, no Cl$_2$ production was observed in Peng et al. when illuminating NaCl solution (Peng et al., 2022), likely due to ineffective photolysis in bulk solution compared to the droplet. Note that Cl$^\bullet$ can also react with S(IV) in the presence of SO$_2$, competing with the bimolecular reaction of Cl$^\bullet$. We used the kinetic model to simulate the effect of SO$_2$ uptake in forming Cl$_2$. As shown in Figure S23, the simulated Cl$_2$ concentration is 3~4 orders of magnitude lower in the presence of

SO$_2$ than in its absence, suggesting that presence of SO$_2$ may inhibit the formation of Cl$_2$. However, Peng et al. suggested that reducing SO$_2$ concentration would slow the Cl$_2$ production due to the reduced particle acidity (Peng et al., 2022). Hence, further systematic studies on such effects are needed in the future.



Recent studies have emphasized the increased contribution of nitrate to $PM_{2.5}$ pollution (Fu et al., 2020; Itahashi et al., 2018; Li et al., 2019a). Thus, identifying the key factors contributing to particulate nitrate formation and driving its trend

is critical to eliminate winter haze episodes. The heterogeneous uptake of $NO_2$ onto aqueous particles, mineral dust, and urban grime with subsequent hydrolysis to form nitrate and HONO has been reported (Dyson et al., 2021; Li et al., 2018a; Liu et al., 2019; Martins-Costa et al., 2020; Pandit et al., 2021; Tan et al., 2016; Xu et al., 2019; Yu et al., 2021). The $\gamma_{NO2}$ is a crucial parameter controlling the heterogeneous chemistry on the aerosol surface in the chemical transport model (McDuffie et al., 2018; Xie et al., 2022). We used the experimentally measured nitrate formation rate from unary uptake of

$NO_2$ into NaCl droplets under dark at all RHs to estimate the $\gamma_{NO2}$ into NaCl droplets in this study. The averaged $\gamma_{NO2}$ are $1.6 \times 10^{-5}$, $1.9 \times 10^{-5}$, and $3.0 \times 10^{-5}$ at 80%, 70%, and 60% RH, respectively. Higher $\gamma_{NO2}$ was observed at low RH, where NaCl droplet achieves supersaturation. Previous studies also reported a much higher uptake coefficient of $NO_2$ of $\sim10^{-4}$ on NaCl particles than other inorganic particles (e.g., $(NH_4)_2SO_4$) (Abbatt and Waschewsky 1998; Ge et al., 2019; Harrison and Collins 1998; Tan et al., 2016), in which the uptake coefficient ranges from $<10^{-8}$ to $10^{-7}$ (Yu et al., 2021). However, Yu et

al. reported a much smaller $\gamma_{NO2}$ on NaCl with the value of $\sim10^{-8}$ (Yu et al., 2021). Existing studies reported a wide range of $\gamma_{NO2}$ (Li et al., 2019b; Miao et al., 2020), bringing difficulties to modelers to estimate the contribution of heterogeneous uptake of $NO_2$ and hydrolysis to form nitrate. For instance, it has been reported that the mean contribution of $NO_2$ hydrolysis to particulate nitrate formation is 6.3~19%, which increases to 35.9% during extreme haze days (Chan et al., 2021; Qiu et al., 2019b; Xie et al., 2022).

In addition, our results also highlighted that presence of $SO_2$ can effectively promote the formation of nitrate further, especially during the nighttime. In other words, controlling $SO_2$ emissions not only reduces sulfate formation, but also slows nitrate formation. Therefore, systematic investigations of the factors, including ionic strength, seed particle types (e.g., nitrate-sulfate-containing particles), and other gas species that impact $NO_2$ uptake and nitrate formation are needed in the future to further constrain the contribution of $NO_2$ hydrolysis to nitrate formation more accurately.






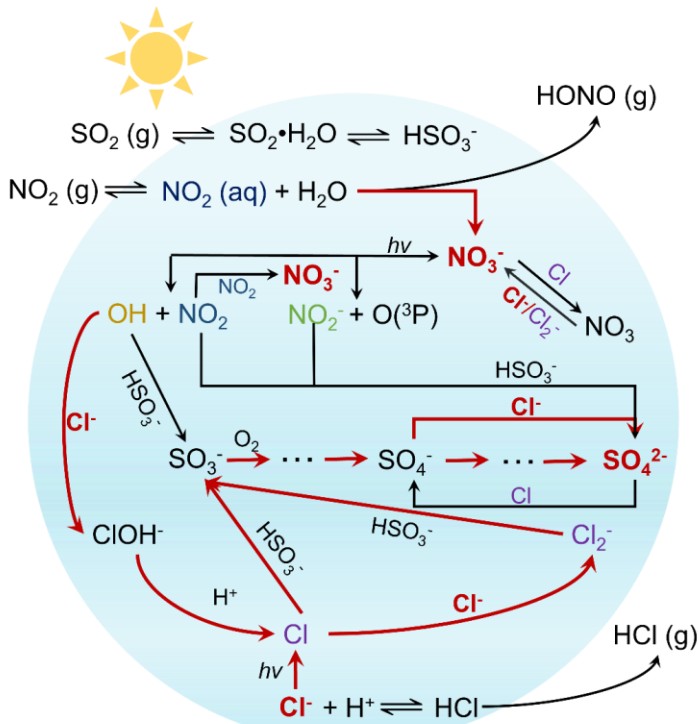

**Figure 1:** The sulfate and nitrate formation mechanisms during the co-uptake of $NO_2$ and $SO_2$ into NaCl droplets under irradiation.



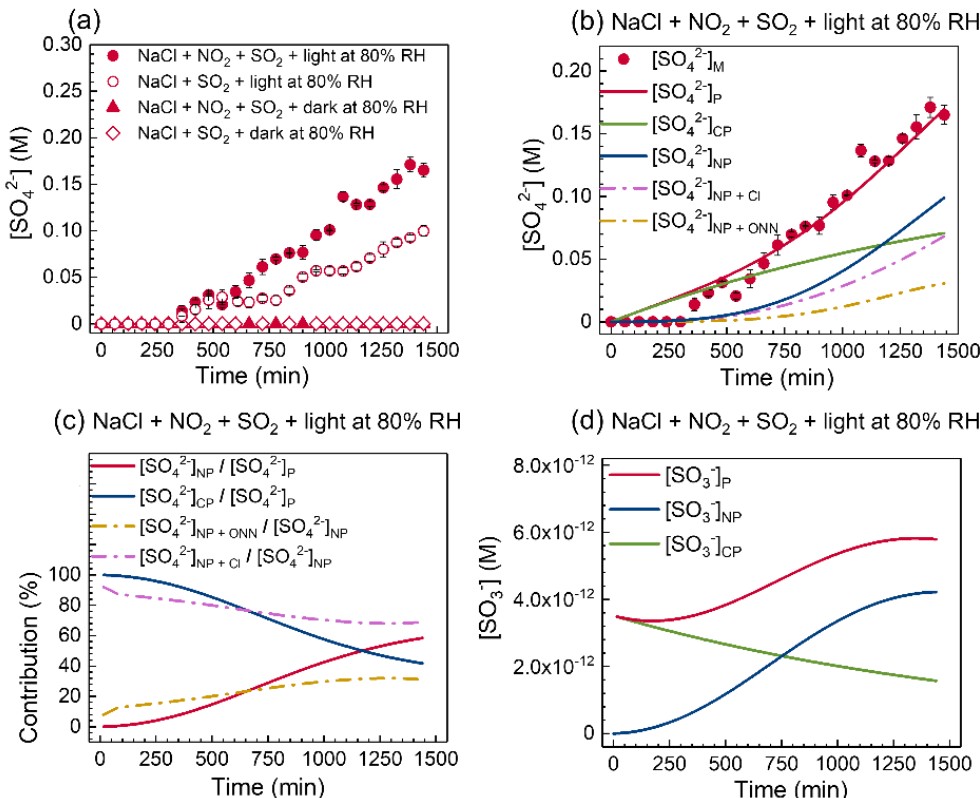

**Figure 2:** (a) Sulfate concentration as a function of time under various conditions, including irradiation/dark experiments and co-/unary uptake experiments at 80% RH. (b) The model predicted sulfate concentration generated from $[SO_4^{2-}]_{NP}$, $[SO_4^{2-}]_{CP}$, $[SO_4^{2-}]_{NP+Cl}$, and $[SO_4^{2-}]_{NP+Cl}$ pathways. (c) The contribution of different formation pathways to sulfate concentration. (d) The model predicted $SO_3^-$ concentration from chloride photolysis and nitrate photolysis. The $[SO_4^{2-}]_M$, $[SO_4^{2-}]_P$, and $[SO_3^-]_P$ represent the experimentally measured $[SO_4^{2-}]$, model-predicted $[SO_4^{2-}]$, and model-predicted $[SO_3^-]$, respectively.



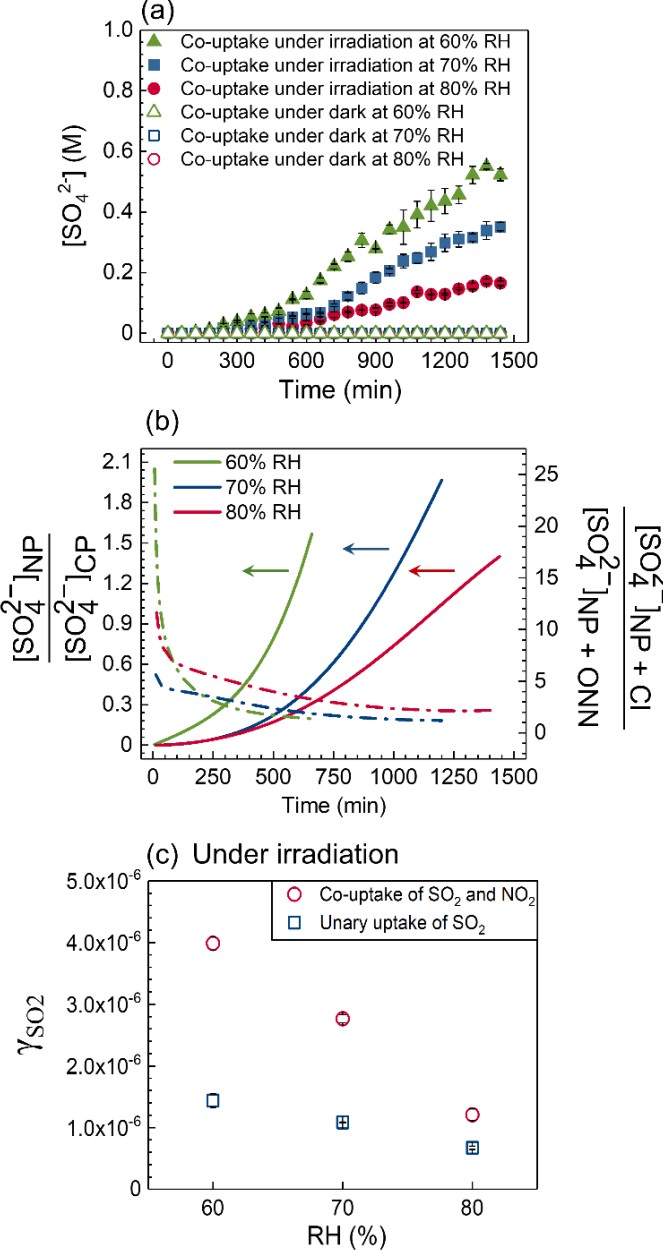

**Figure 3:** (a) Sulfate concentration as a function of time during the co-uptake of $NO_2$ and $SO_2$ into NaCl droplets at 60%, 70%, and 80% RH under dark and irradiation. (b) $[SO_4^{2-}]_{NP}$ / $[SO_4^{2-}]_{CP}$ and $[SO_4^{2-}]_{NP + Cl}$ / $[SO_4^{2-}]_{NP + ONN}$ as a function of time at 80%, 70%, and 60% RH. The dash lines in panel (b) refer to the right y axis. (c) Reactive uptake coefficient of $SO_2$, $\gamma_{SO2}$, at different RHs under irradiation during co-uptake and unary uptake.



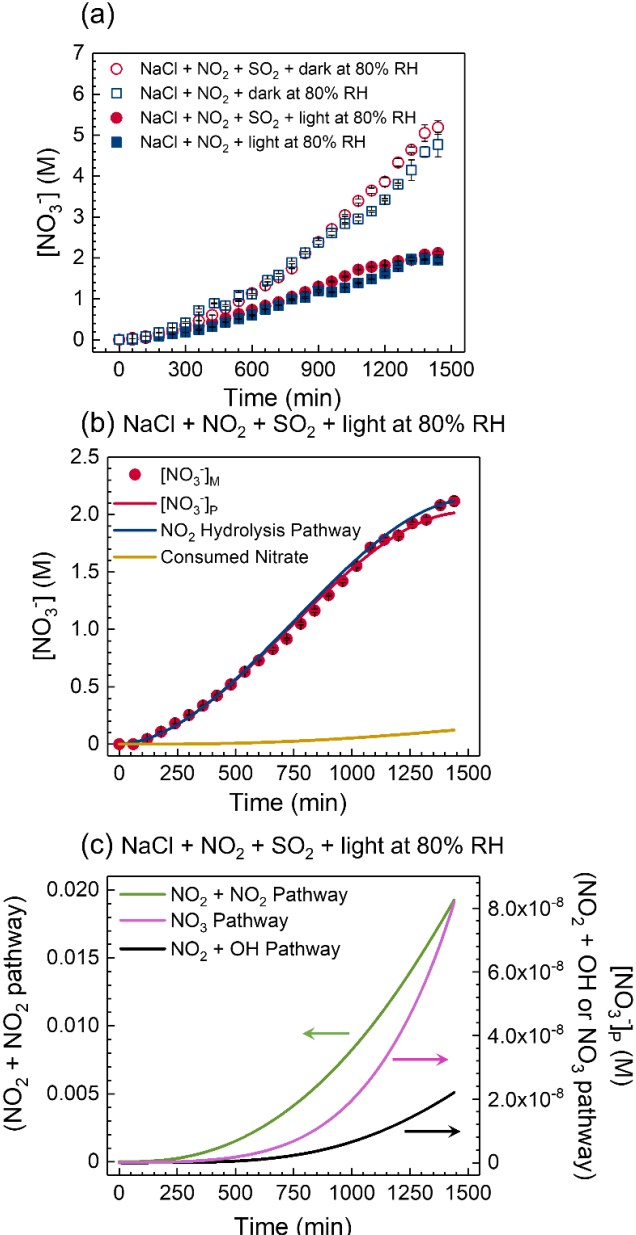

**Figure 4:** (a) The nitrate concentration as a function of time in co-/unary uptake experiments under irradiation/dark conditions at 80% RH. (b) and (c) The experimentally measured nitrate concentration, $[NO_3^-]_M$, and model predicted nitrate concentration, $[NO_3^-]_P$, from different pathways during the co-uptake of $NO_2$ and $SO_2$ under irradiation at 80% RH. "Consumed Nitrate" refers to the reduction in nitrate concentration due to nitrate photolysis (i.e., SR1 in Table S1).





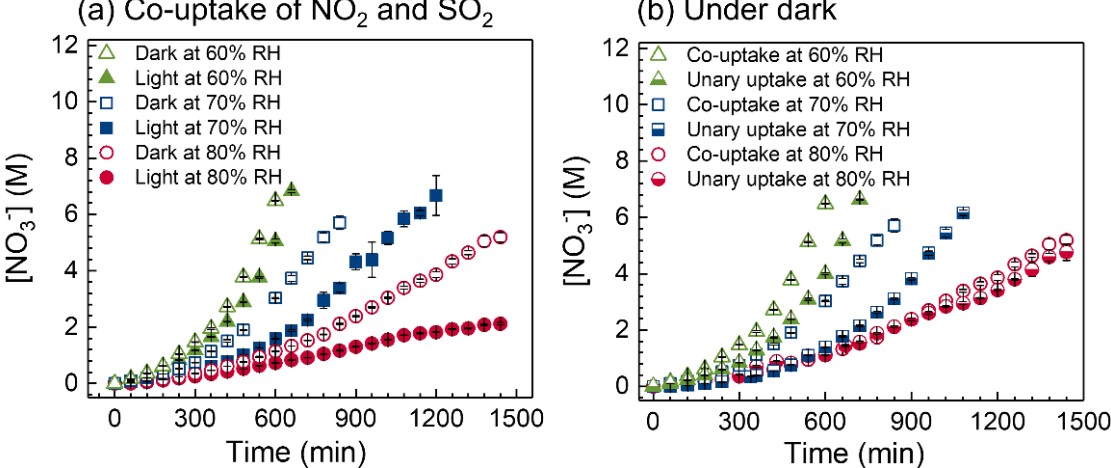

**Figure 5:** (a) Nitrate concentration as a function of time during the co-uptake of $NO_2$ and $SO_2$ into NaCl droplets at different RHs under dark and irradiation. (b) Nitrate concentration as a function of time under various conditions, including co-uptake of $NO_2$ and $SO_2$ and unary uptake of $NO_2$ under dark at 60%, 70%, and 80% RH.

**Data availability.** All data used in this study are available in public archives.

**Author contributions.** R. Z. and C. K. C designed the whole research. R. Z. conducted experiments and kinetic model simulation. R. Z. and C. K. C. analyzed the experimental data and wrote the paper.

**Competing interests.** The authors declare no competing interest.

**Financial support.** This work was supported by the National Natural Science Foundation of China (No. 42075100, 41875142, and 42275104), the Guangdong Basic and Applied Basic Research Foundation (2020B1515130003), and Hong Kong Research Grants Council (11304121).



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
