# Peer review of "Simultaneous Formation of Sulfate and Nitrate via Co-uptake of SO2 and NO2 by Aqueous NaCl Droplets: Combined Effect of Nitrate Photolysis and Chlorine Chemistry"

_EGUsphere, 2023_

## Author Comment (AC1)

**Reviewer #1**

*The study by Zhang et al. reported sulfate and nitrate formation during the co-uptake of NO2 and SO2 into NaCl droplets with chamber experiments and kinetic modelling. In general, the study is well designed and presented, and I'd recommend the publishment if the following issues can be well addressed.*

**Response:**

We thank the reviewer for the valuable comments. We have revised the manuscript according to the questions raised by the reviewer.

***Comment 1.*** *Figure 1: What does the red and black arrows mean? In the OH+NO2 to NO3- under hv reaction, why there're arrows on both ends? The arrow on the NO3- side should be removed if it's indicating SR1.*

**Response:**

Figure 1 shows the overall reaction mechanisms during $NO_2$ and $SO_2$ uptake into NaCl droplets under irradiation. We highlighted the interaction of nitrate photolysis and chlorine chemistry in forming sulfate in red arrows. To eliminate the potential confusion, we have revised the caption of Figure 1.

Nitrate photolysis generates OH radicals and $NO_2$, and the reaction of OH radicals with $NO_2$ can regenerate the nitrate. We have also revised Figure 1 by removing the arrow on the $NO_3^-$ side and adding the separate arrows from OH + $NO_2$ to $NO_3^-$ and $NO_2^- + O(^3P)$ to $NO_3^-$.

**Revision in the manuscript (Figure 1):**

[Figure]

Figure 1. The sulfate and nitrate formation mechanisms during the co-uptake of $NO_2$ and $SO_2$ into NaCl droplets under irradiation. The red arrows highlight the role of enhanced chlorine chemistry triggered by OH radicals produced from nitrate photolysis in forming sulfate.

**Comment 2.** *Section 3.2: The concentrations of which oxidants increase at low RH? And does the rate constant change?*

**Response:**

During unary uptake of $SO_2$ into NaCl droplets, the primary oxidant of Cl radicals generated from chloride photolysis increase due to the increased chloride concentration at low RH.

During the co-uptake of $SO_2$ and $NO_2$ into NaCl droplets, the oxidants of reactive species generated from nitrate photolysis (i.e., OH radical, $NO_2$, and N(III)) and Cl radical from chlorine chemistry all increase at low RH.

The RH does not affect the rate constant. However, low RH, that is, low water content, would increase the reactant concentration, accelerating the reaction rate subsequently.

**Revision in the manuscript (Section 3.2, Line 205-210):**

"As shown in Figure 3a and Figure S11, the sulfate production rate (with unit of $M \cdot s^{-1}$), $d[SO_4^{2-}]/dt$, increases with decreasing RH, which is attributed to the increased concentrations of oxidants (i.e., OH radical, $NO_2$, N(III), and $Cl^\bullet$ /$Cl_2^{\bullet-}$) at low RH, irrespective of the presence of $NO_2$ or not. It should be noted that the total sulfate production rate (with unit of $mole \cdot s^{-1}$), $d(n(SO_4^{2-}))/dt$, depends on the size of the droplets. However, the comparable droplet size utilized in all experiments implies $d(n(SO_4^{2-}))/dt$ follows the same trends as the $d[SO_4^{2-}]/dt$."

**Comment 3.** *Line 243-244: If the derived expression is "constrained to the conditions in the presence of nitrate and chloride", I'd expect a range of the corresponding $[NO_3^-]/[Cl^-]$ ratio, not only an upper limit (i.e.< 3). Moreover, is there any constraint on the relative ratio to $SO_2/NO_2$? A systematic modelling study would help if the models have been validated by the experimental results.*

**Response:**

We constrained the $[NO_3^-]/[Cl^-]$ to below 3 because this is the concentration range we studied. We are cautious not to extrapolate our findings above this concentration ratio because we don't have sufficient experimental evidence. Future work is needed to explore whether this expression applies to the large range of

$[NO_3^-]/[Cl^-]$.

In our study, we used a fixed ratio of $SO_2/NO_2$ (6.5 ppm / 10 ppm) to study the sulfate and nitrate formation mechanisms during the $NO_2$ and $SO_2$ uptake into NaCl droplets under irradiation. Moreover, we have demonstrated that the derived expression is independent of the ratio of $SO_2$ to $NO_2$ (see the response to comment #6 below). Future work with an atmospherically relevant ratio of $SO_2/NO_2$ and incorporation of these reactions in chemical transport models would be useful.

**Revision in the manuscript (Section 3.2, Line 240-245):**

"The derived expression matched with the experimental data well (Figure S14). Our experimental results pertain to $[NO_3^-]/[Cl^-]$ below 3, and the derived equation applies to the system involving nitrate and chloride both. Extrapolation beyond this range should be done with caution."

*Comment 4. Line 299: How can the reaction of NO2 with S(IV) form nitrate in droplets under dark conditions? Isn't it nitrite?*

**Response:**

Thank you for your careful reading. Yes, the reaction of $NO_2$ with S(IV) should yield nitrite but not nitrate. We meant to say that $NO_2$ hydrolysis and the reaction of $NO_2$ with S(IV) are the only two reactions during the co-uptake of $NO_2$ and $SO_2$ under dark. Sorry for the mistakes. We have revised the corresponding texts in line 299.

**Revision in the manuscript (Section 3.4, Line 299):**

Under dark, $NO_2$ hydrolysis and the reaction of $NO_2$ with S(IV) are the only two reactions  occurring in droplets, with the former  being the sole pathway in forming nitrate.

*Comment 5. While the aqueous-phase sulfate production rate increase with decreasing RH, how about the total sulfate production rate? That is, will the increased rate overcome the decreased droplet water and result in an overall more sulfate production rate with decreasing RH?*

**Response:**

We assumed that the "total sulfate production rate" the reviewer referred is the $d(n(SO_4^{2-}))/dt$ (mole/s). The $n(SO_4^{2-})$ can be expressed as:

$$n(SO_4^{2-}) = [SO_4^{2-}] \times V_{droplet},$$

where $n(SO_4^{2-})$ and $[SO_4^{2-}]$ are the total moles of sulfate (mole) and sulfate concentration (mole/L) in droplets, respectively. $V_{droplet}$ is the droplet volume. Because

the droplet size ((57±2) μm) used among different conditions are comparable, the calculated $n(SO_4^{2-})$ should be proportional to $[SO_4^{2-}]$. Thus, the absolute total sulfate production rate also increases as RH decreases. We have added the following texts to address this issue.

**Revision in the manuscript (Section 3.2, Line 205-210):**

"As shown in Figure 3a and Figure S11, the sulfate production rate (with unit of $M \cdot s^{-1}$), $d[SO_4^{2-}]/dt$, increases with decreasing RH, which is attributed to the increased concentrations of oxidants (i.e., OH radical, $NO_2$, N(III), and $Cl^\bullet$ /$Cl_2^{\bullet-}$) at low RH, irrespective of the presence of $NO_2$ or not. It should be noted that the total sulfate production rate (with unit of $mole \cdot s^{-1}$), $d(n(SO_4^{2-}))/dt$, depends on the size of the droplets. However, the comparable droplet size utilized in all experiments implies $d(n(SO_4^{2-}))/dt$ follows the same trends as the $d[SO_4^{2-}]/dt$."

*Comment 6. Section 4: I feel this part is extending too much. For urban aerosols, the major composition is usually ammonium sulfate and barely NaCl, and the whole pH range would differ. I'd suggest revise this part and focus mainly on the implications on sea salt chemistry. In addtion, with the parameterization given (i.e., γSO2= 0.41 × PNO3- + 0.34 × PCl-), how much sulfate can be produced in the atmosphere under typical coastal conditions?*

**Response:**

While ammonium sulfate may be more dominant inland, NaCl is an important PM component in many coastal cities. It is well known that nitrate and chloride coexist in sea salt aerosols due to the chloride depletion reactions (e.g., Zhuang et al., 1999ab).

Furthermore, although the current work focuses on the NaCl droplets, the proposed mechanism for the interaction of nitrate photolysis and chlorine chemistry in forming sulfate is not restricted to sea salt aerosol. In the inland environment, chloride and nitrate may predominantly exist as ammonium salts. Hence, we performed additional experiments with $SO_2$ uptake into pure $NH_4NO_3$ droplets and premixed $NH_4Cl + NH_4NO_3$ (4:1 in mole ratio at 75% RH) under irradiation (Xe lamp). Figure Aa shows a higher sulfate production rate in pure $NH_4NO_3$ droplets, which is likely due to the higher initial nitrate concentration in pure $NH_4NO_3$ droplets (~8.1M) than in premixed $NH_4Cl + NH_4NO_3$ droplets (~1.0 M). As reported in our earlier works, nitrate photolysis could also promote sulfate formation (Gen et al., 2019). Hence, sulfate concentration is normalized by the initial nitrate concentration, as shown in Figure Ab. A significant enhancement in the normalized sulfate production rate was observed in premixed $NH_4Cl + NH_4NO_3$ droplets, implying that the interplay of nitrate photolysis and chlorine chemistry could also take place. In addition, as mentioned in section 4 in the main text (Line 345-355), our previous work with $SO_2$ uptake into premixed $NH_4Cl + NH_4NO_3$ droplet under 300-nm irradiation also supports that our proposed mechanisms are not only limited to sea salt particles (Zhang et al., 2020).

Note that the derived expression (Equation 1) is based on the experimental conditions with co-uptake of $NO_2$ and $SO_2$ into NaCl droplets under irradiation at 60%, 70%, and 80% RH.

$$\gamma_{SO2} = 0.41 \times P_{NO3^-} + 0.34 \times P_{Cl^-} \qquad \text{(Equation 1)}$$

To evaluate if the equation is applicable to non sea salt system, we compared $\gamma_{SO2}$ from the Equation 1 with that from experimental results (i.e., premixed $NH_4Cl$ + $NH_4NO_3$ system) in Figure Aa. A kinetic model (FACSIMILE software) was used to obtain the fitted nitrate and chloride photolysis rate constants in premixed $NH_4Cl$ + $NH_4NO_3$ system, which are $1.0 \times 10^{-6}$ s$^{-1}$ and $1.6 \times 10^{-6}$ s$^{-1}$, respectively. In experiments of premixed $NH_4Cl$ + $NH_4NO_3$ droplets equilibrated at 75% RH, the initial concentration of nitrate and chloride are 1.0 M and 4.0 M, respectively. Hence, the nitrate photolysis rate, $P_{NO3^-}$, and chloride photolysis rate, $P_{Cl^-}$, are the $1.0 \times 10^{-6}$ M s$^{-1}$ and $6.5 \times 10^{-6}$ M s$^{-1}$, respectively. The calculated $\gamma_{SO2}$ ($2.6 \times 10^{-6}$) agrees well with that from the experiments ($3.0 \times 10^{-6}$) (Text S2 in Supporting Information). This gives us confidence that the derived expression of uptake coefficient may be useful in ammonium system containing nitrate and chloride too.

Based on our proposed mechanisms and expression for the uptake coefficient, we estimated the sulfate production rate using characteristic data from Beijing reported by Huang et al. (2014). The reported concentrations of nitrate, sulfate, chloride, ammonium, and OM are summarized in Table A. The E-AIM was used to estimate the contribution of all inorganic components to aerosol water content (AWC), $W_{inorganic}$, by assuming RH of 60% during the haze events. The contribution of OM to AWC, $W_{org}$, was estimated by the same approach of Guo et al. (2015).

$$W_{org} = \frac{OM}{\rho_{org}} \cdot \rho_w \cdot \frac{k_{org}}{(\frac{100\%}{RH}-1)} \qquad \text{(Equation 2)}$$

where OM is the mass concentration of organic matter, $\rho_w$ is the density of water ($\rho_w$= $1.0 \times 10^3$ kg m$^{-3}$), $\rho_{org}$ is the density of organics ($\rho_{org}$= $1.4 \times 10^3$ kg m$^{-3}$), and $k_{org}$ is the hygroscopicity parameter of organic aerosol composition. We adopted a $k_{org}$ of 0.06 based on previous cloud condensation nuclei measurements in Beijing. Overall, the estimated total AWC ($=W_{inorganic} + W_{org}$) taking into contributions by the inorganics and OM is ~46 µg m$^{-3}$.

The sulfate production rate in unit of µg m$^{-3}$ h$^{-1}$, $\frac{d[SO_4^{2-}]}{dt}$, can be calculated by:

$$\frac{d[SO_4^{2-}]}{dt} = 3600 \, s \, h^{-1} \times 10^{-3} \times M(SO_4^{2-}) \times R_{SO42-} \times \frac{AWC}{\rho_w} \qquad \text{(Equation 3)}$$

where $M(SO_4^{2-})$ is 96.06 g mol$^{-1}$, AWC (~46 µg m$^{-3}$) and $\rho_w$ (=1 kg L$^{-1}$) are aerosol water content and water density (Liu et al., 2021), respectively. $R_{SO42-}$ is the sulfate production rate in unit of M s$^{-1}$, which can be obtained by the uptake coefficient:

$$R_{SO42-} = \frac{1}{4}\omega[SO_2]\gamma_{SO2}A_s \times 10^{-3} \qquad \text{(Equation 4)}$$

where [$SO_2$] (40 ppb) is the gas phase concentration of $SO_2$ (mole m$^{-3}$ air), $\omega$ (=

$(8RT/\pi M)^{1/2}$) is the mean speed of the molecule, $A$s ($=3/r_P$) is the surface area of a particle per volume ($r_P = 0.15$ μm) (Cheng et al., 2016).

According to previous studies (Ye et al., 2017, Gen et al., 2019), particulate nitrate photolysis rate constant under >290 nm irradiation can achieve ~$10^{-5}$ s$^{-1}$. The chloride photolysis rate constant is rarely reported in the literature; hence, we used the averaged photolysis rate constant obtained from fitting in our investigation (Table S3 in Supporting Information), which was on the order of ~$10^{-7}$ s$^{-1}$. Based on these values, we estimated $\gamma_{SO2}$ to be ~$2.7 \times 10^{-5}$, calculated from Equation 1. Using Equations 3 and 4 with these values, the sulfate production rate is estimated to be 1.1 μg m$^{-3}$ h$^{-1}$, which is comparable to the sulfate production rates from other traditionally reported pathways (e.g., $O_3$, TMI, $H_2O_2$, and $NO_2$ pathways) at particle pH of 4-6 during haze events (Liu et al., 2021). Future chemical model simulation of the contribution of proposed mechanisms to sulfate production rate is needed to evaluate its impact in the atmosphere thoroughly.

**Table A.** The chemical composition of collected PM$_{2.5}$ during the high pollution events of 5–25 January 2013 at the urban sites of Beijing (Huang et al., 2014).

| Component | nitrate | chloride | sulfate | ammonium | Organic compounds (OM) |
|---|---|---|---|---|---|
| Mass Concentration (μg m$^{-3}$ air) | 19.0 | 6.2 | 25.4 | 15.5 | 64.5 |
| Mole Concentration (μmole m$^{-3}$ air) | 0.3 | 0.17 | 0.3 | 0.9 | / |
| Molar concentration (M) | 6.5 | 3.7 | 6.5 | 19.6 | / |
| Water content ($\mu$g m$^{-3}$ air) | $W_{inorganic}$: 41.8 | | | | $W_{org}$: 4.1 |

**Revision in the manuscript (section 4):**

The present study indicated a previously unrecognized role of the combined effect of nitrate photolysis and chlorine chemistry in enhanced sulfate formation, which is expected to have important implications on atmospheric chemistry. In brief, the OH radicals formed from nitrate photolysis can react with chloride ions to promote the formation of chlorine species (e.g., Cl$^{\bullet}$ and Cl$_2^{\bullet-}$) and further enhance sulfate formation. Such effect enhances the $\gamma_{SO2}$ by factors of ~1.7, ~2.5, and ~2.9 at 80%, 70%, and 60% RH at 0.2 < [NO$_3^-$]/[Cl$^-$] < 3, respectively, compared to those from the chloride photolysis (i.e., NaCl + SO$_2$ + light conditions).

To further verify whether our proposed mechanism could apply urban sites, where ammonium salts are the major components of particles, we performed additional experiments with SO$_2$ uptake into premixed NH$_4$Cl + NH$_4$NO$_3$ (4:1 in mole ratio) droplets and pure NH$_4$Cl droplets under the same irradiation at 75% RH. Figure S23a shows a higher sulfate production rate in pure NH$_4$NO$_3$ droplets, likely due to the higher

initial nitrate concentration in pure $NH_4NO_3$ droplets (~8.1 M) than in premixed $NH_4Cl$ + $NH_4NO_3$ droplets (~1.0 M). As reported in our earlier works, nitrate photolysis could also promote sulfate formation. Hence, sulfate concentration is normalized by initial nitrate concentration, as shown in Figure S23b. A significant enhancement in the normalized sulfate production rate was observed in premixed $NH_4Cl$ + $NH_4NO_3$ droplets, confirming that the interplay of nitrate photolysis and chlorine chemistry could also take place. Note that the uptake coefficient equation ($\gamma_{SO2} = 0.41 \times P_{NO3-}$ + $0.34 \times P_{Cl-}$) is based on the experimental results of the co-uptake of $NO_2$ and $SO_2$ into NaCl droplets under irradiation at 60%, 70%, and 80% RH. To evaluate if this equation is applicable to non sea salt system, we compared $\gamma_{SO2}$ from the equation with that from experimental results (i.e., premixed $NH_4Cl$ + $NH_4NO_3$ system) in Figure S23a . The calculated $\gamma_{SO2}$ ($2.6 \times 10^{-6}$) agrees well with that from the experiments ($3.0 \times 10^{-6}$) (Text S2). This gives us confidence that the derived expression of uptake coefficient may be useful in ammonium system containing nitrate and chloride too. In our earlier work, we performed unary uptake of $SO_2$ into premixed $NH_4Cl$ + $NH_4NO_3$ droplets under 300-nm irradiation at different $[NO_3^-]/[Cl^-]$, in air and $N_2$ environment (Zhang et al., 2020). Note that the presence of $O_2$ (air environment) is essential for sulfate formation from reactions of reactive chlorine species with $HSO_3^-$ (Figure 1). The normalized sulfate production rate continuously increases as $[NO_3^-]/[Cl^-]$ decreases in air, but no further enhancement of sulfate production rate was observed in $N_2$ when $[NO_3^-]/[Cl^-]<5$ (Zhang et al., 2020). Such results also support the combined effect of nitrate photolysis and chlorine chemistry in promoting sulfate formation may be applicable in ammonium systems containing nitrate and chloride.

[Figure]

**Figure A.** (a) Sulfate concentration as a function of time in premixed $NH_4Cl$ + $NH_4NO_3$ (4:1 in mole ratio) and pure $NH_4NO_3$ droplets under irradiation at 75% RH. (b) Sulfate concentration normalized by the initial nitrate concentration under the same conditions as panel (a).

Reference:

(1) Zhang, R.; Gen, M.; Huang, D.; Li, Y.; Chan, C. K. Enhanced Sulfate Production

by Nitrate Photolysis in the Presence of Halide Ions in Atmospheric Particles. *Environ. Sci. Technol.* **2020,** *54* (7), 3831-3839.

(2) Cheng, Y.; Zheng, G.; Wei, C.; Mu, Q.; Zheng, B.; Wang, Z.; Gao, M.; Zhang, Q.; He, K.; Carmichael, G.; Pöschl, U.; Su, H., Reactive nitrogen chemistry in aerosol water as a source of sulfate during haze events in China. *Sci. Adv.* **2016**, 2, (12).e1601530

(3) Ye, C.; Zhang, N.; Gao, H.; Zhou, X. Photolysis of Particulate Nitrate as a Source of HONO and NOx. *Environ. Sci. Technol.* **2017**, 51 (12), 6849−6856.

(4) Gen, M.; Zhang, R.; Huang, D. D.; Li, Y.; Chan, C. K. Heterogeneous Oxidation of SO2 in Sulfate Production during Nitrate Photolysis at 300 nm: Effect of pH, Relative Humidity, Irradiation Intensity, and the Presence of Organic Compounds. *Environ. Sci. Technol.* **2019,** *53* (15), 8757-8766.

(5) H. Guo, L. Xu, A. Bougiatioti, K. Cerully, S. Capps, J. Hite, A. Carlton, S. Lee, M. Bergin, N. Ng, A. Nenes, R. Weber, Fine-particle water and pH in the southeastern United States. *Atmos. Chem. Phys.* 15, 5211–5228 (**2015**).

(6) Huang, R.-J.; Zhang, Y.; Bozzetti, C.; Ho, K.-F.; Cao, J.-J.; Han, Y.; Daellenbach, K. R.; Slowik, J. G.; Platt, S. M.; Canonaco, F.; Zotter, P.; Wolf, R.; Pieber, S. M.; Bruns, E. A.; Crippa, M.; Ciarelli, G.; Piazzalunga, A.; Schwikowski, M.; Abbaszade, G.; Schnelle-Kreis, J.; Zimmermann, R.; An, Z.; Szidat, S.; Baltensperger, U.; Haddad, I. E.; Prévôt, A. S. H. High secondary aerosol contribution to particulate pollution during haze events in China. *Nature* **2014,** *514* (7521), 218-222.

(7) Zhuang, H., Chan, C. K., Fang, M., and Wexler, A. S.: Size distributions of particulate sulfate, nitrate, and ammonium at a coastal site in Hong Kong, *Atmos. Environ.,* 33, 843-853, 1999a.

(8) Zhuang, H., Chan, C. K., Fang, M., and Wexler, A. S.: Formation of nitrate and non-sea-salt sulfate on coarse particles, *Atmos. Environ.,* 33, 4223-4233, 1999b.

*Comment 7. There're some typos and misuse of sysbols, and the authors should check carefully. For example, there's a blank line in the end of Table S1. Is there any reaction missing or simply a typo?*

**Response:**

Thank you for your careful review. We have done a thorough check to correct the potential typos. There is no missing reaction in Table S1, and we have deleted the blank line.

---

## Author Comment (AC2)

**Reviewer #2**

*The manuscript by Zhang and Chan reports an experimental result for reactive uptake of SO2 and NO2 by aqueous NaCl droplets using the micro Raman spectroscopy. The experiment was conducted both under dark and irradiation by a Xe lamp. Nitrate formation was observed for droplets both under dark and light-irradiated conditions. Sulfate was observed only when light was available. The experimental result was parameterized using a chemical kinetic model.*

*Both the experimental technique and result sound. The major conclusion is well supported by the experimental data. The topic is within the scope of the interest of the readers of the journal, although the reviewer wonders if the concentration ranges of chemical species and light intensities are atmospheric relevant. The manuscript is reasonably well written. Revision will be required prior to the publication of this manuscript.*

**Response:**

We thank the reviewer for the valuable comments. We have revised the manuscript according to the questions raised by the reviewer.

*Major comments*

**Comment 1.** *Atmospheric relevance of the experimental conditions*

*The concentration ranges for both SO2 (~6.5 ppm) and NO2 (~10 ppm) were a few orders higher than that for typical atmospheric conditions. As the authors constructed a chemical kinetic model for reproducing the experimental result, it might be interesting to simulate if the finding of the present study would be important for atmospheric relevant concentration and light intensity ranges.*

**Response:**

We agree that such concentrations are not atmospherically relevant. To our best knowledge, this is probably the first experimental investigation of the mechanisms of nitrate and sulfate formation during the $NO_2$ and $SO_2$ uptake into NaCl seed particles under irradiation. In this study, we aimed to have a better understanding of their formation mechanism, especially in the presence of irradiation. Using high concentrations of $SO_2$ and $NO_2$ allowed the formation of nitrate and sulfate in a reasonable time to determine their respective formation rate.

Although the derived expression (Equation 1) was from the experimental conditions with co-uptake of $NO_2$ and $SO_2$ into NaCl droplets under irradiation at 60%, 70%, and 80% RH, it is based on the nitrate and chloride concentration in the droplets; hence, in principle, it is independent of the $SO_2$ and $NO_2$ concentrations and seed particle types.

$$\gamma_{SO2} = 0.41 \times P_{NO3-} + 0.34 \times P_{Cl-} \qquad \text{(Equation 1)}$$

To verify it, we performed additional experiments with $SO_2$ uptake into premixed

$NH_4Cl + NH_4NO_3$ droplets (4:1 in mole ratio at 75% RH) and pure $NH_4Cl$ droplets under irradiation (Xe lamp), in which no $NO_2$ gas, but the interaction of nitrate photolysis and chlorine chemistry still can occur in premixed system. Figure Aa shows a higher sulfate production rate in pure $NH_4NO_3$ droplets, which is likely due to the higher initial nitrate concentration in pure $NH_4NO_3$ droplets (~8.1M) than in premixed $NH_4Cl + NH_4NO_3$ droplets (~1.0 M). As reported in our earlier works, nitrate photolysis could also promote sulfate formation. Hence, sulfate concentration is normalized by initial nitrate concentration, as shown in Figure Ab. A significant enhancement in the normalized sulfate production rate was observed in premixed $NH_4Cl + NH_4NO_3$ droplets, implying that the interplay of nitrate photolysis and chlorine chemistry could also take place.

To evaluate if the equation is applicable to non sea salt system, we compared $\gamma_{SO2}$ from the Equation 1 with that from experimental results (i.e., premixed $NH_4Cl + NH_4NO_3$ system) in Figure Aa. A kinetic model (FACSIMILE software) was used to obtain the fitted nitrate and chloride photolysis rate constants in premixed $NH_4Cl + NH_4NO_3$ system, which are $1.0 \times 10^{-6}$ s$^{-1}$ and $1.6 \times 10^{-6}$ s$^{-1}$, respectively. In experiments of premixed $NH_4Cl + NH_4NO_3$ droplets equilibrated at 75% RH, the initial concentration of nitrate and chloride are 1.0 M and 4.0 M, respectively. Hence, the nitrate photolysis rate, $P_{NO3-}$, and chloride photolysis rate, $P_{Cl-}$, are the $1.0 \times 10^{-6}$ M s$^{-1}$ and $6.5 \times 10^{-6}$ M s$^{-1}$, respectively. The calculated $\gamma_{SO2}$ ($2.6 \times 10^{-6}$) agrees well with that from the experiments ($3.0 \times 10^{-6}$) (Text S2 in Supporting Information). This gives us confidence that the derived expression of uptake coefficient may be useful in ammonium system containing nitrate and chloride too.

Based on our proposed mechanisms and expression for the uptake coefficient, we estimated the sulfate production rate using characteristic data from Beijing reported by Huang et al. (2014). The reported concentrations of nitrate, sulfate, chloride, ammonium, and OM are summarized in Table A. The E-AIM was used to estimate the contribution of all inorganic components to aerosol water content (AWC), $W_{inorganic}$, by assuming RH of 60% during the haze events. The contribution of OM to AWC, $W_{org}$, was estimated by the same approach of Guo et al. (2015).

$$W_{org} = \frac{OM}{\rho_{org}} \cdot \rho_w \cdot \frac{k_{org}}{(\frac{100\%}{RH}-1)} \qquad \text{(Equation 2)}$$

where OM is the mass concentration of organic matter, $\rho_w$ is the density of water ($\rho_w = 1.0 \times 10^3$ kg m$^{-3}$), $\rho_{org}$ is the density of organics ($\rho_{org} = 1.4 \times 10^3$ kg m$^{-3}$), and $k_{org}$ is the hygroscopicity parameter of organic aerosol composition. We adopted a $k_{org}$ of 0.06 based on previous cloud condensation nuclei measurements in Beijing. Overall, the estimated total AWC ($= W_{inorganic} + W_{org}$) taking into contributions by the inorganics and OM is ~46 μg m$^{-3}$.

The sulfate production rate in unit of μg m$^{-3}$ h$^{-1}$, $\frac{d[SO_4^{2-}]}{dt}$, can be calculated by:

$$\frac{d[SO_4^{2-}]}{dt} = 3600 \, s \, h^{-1} \times 10^{-3} \times M(SO_4^{2-}) \times R_{SO42-} \times \frac{AWC}{\rho_w} \qquad \text{(Equation 3)}$$

where $M(SO_4^{2-})$ is 96.06 g mol$^{-1}$, AWC (~46 μg m$^{-3}$) and $\rho_w$ (=1 kg L$^{-1}$) are aerosol water content and water density (Liu et al., 2021), respectively. $R_{SO42-}$ is the sulfate production rate in unit of M s$^{-1}$, which can be obtained by the uptake coefficient:

$$R_{SO42-} = \frac{1}{4}\omega[SO_2]\gamma_{SO2}A_s \times 10^{-3} \qquad \text{(Equation 4)}$$

where [SO$_2$] (40 ppb) is the gas phase concentration of SO$_2$ (mole m$^{-3}$ air), $\omega$ (= $(8RT/\pi M)^{1/2}$) is the mean speed of the molecule, $As$ (=3/r$_p$) is the surface area of a particle per volume (r$_p$ = 0.15 μm) (Cheng et al., 2016).

According to previous studies (Ye et al., 2017, Gen et al., 2019), particulate nitrate photolysis rate constant under >290 nm irradiation can achieve ~10$^{-5}$ s$^{-1}$. The chloride photolysis rate constant is rarely reported in the literature; hence, we used the averaged photolysis rate constant obtained from fitting in our investigation (Table S3 in Supporting Information), which was on the order of ~10$^{-7}$ s$^{-1}$. Based on these values, we estimated $\gamma_{SO2}$ to be ~2.7 × 10$^{-5}$, calculated from Equation 1. Using Equations 3 and 4 with these values, the sulfate production rate is estimated to be 1.1 μg m$^{-3}$ h$^{-1}$, which is comparable to the sulfate production rates from other traditionally reported pathways (e.g., O$_3$, TMI, H$_2$O$_2$, and NO$_2$ pathways) at particle pH of 4-6 during haze events (Liu et al., 2021). Future chemical model simulation of the contribution of proposed mechanisms to sulfate production rate is needed to evaluate its impact in the atmosphere thoroughly.

**Table A.** The chemical composition of collected PM$_{2.5}$ during the high pollution events of 5–25 January 2013 at the urban sites of Beijing (Huang et al., 2014).

| Component | nitrate | chloride | sulfate | ammonium | Organic compounds (OM) |
|---|---|---|---|---|---|
| Mass Concentration (μg m$^{-3}$ air) | 19.0 | 6.2 | 25.4 | 15.5 | 64.5 |
| Mole Concentration (μmole m$^{-3}$ air) | 0.3 | 0.17 | 0.3 | 0.9 | / |
| Molar concentration (M) | 6.5 | 3.7 | 6.5 | 19.6 | / |
| Water content (μg m$^{-3}$ air) | $W_{inorganic}$: 41.8 | | | | $W_{org}$: 4.1 |

**Revision in the manuscript (section 4):**

[revised manuscript text omitted]

(5) H. Guo, L. Xu, A. Bougiatioti, K. Cerully, S. Capps, J. Hite, A. Carlton, S. Lee, M. Bergin, N. Ng, A. Nenes, R. Weber, Fine-particle water and pH in the southeastern United States. *Atmos. Chem. Phys.* 15, 5211–5228 (**2015**).

(6) Huang, R.-J.;  Zhang, Y.;  Bozzetti, C.;  Ho, K.-F.;  Cao, J.-J.;  Han, Y.;  Daellenbach, K. R.;  Slowik, J. G.;  Platt, S. M.;  Canonaco, F.;  Zotter, P.;  Wolf, R.;  Pieber, S. M.; Bruns, E. A.;  Crippa, M.;  Ciarelli, G.;  Piazzalunga, A.;  Schwikowski, M.;  Abbaszade, G.;  Schnelle-Kreis, J.;  Zimmermann, R.;  An, Z.;  Szidat, S.;  Baltensperger, U.; Haddad, I. E.; Prévôt, A. S. H. High secondary aerosol contribution to particulate pollution during haze events in China. *Nature* **2014,** *514* (7521), 218-222.

(7) Zhuang, H., Chan, C. K., Fang, M., and Wexler, A. S.: Size distributions of particulate sulfate, nitrate, and ammonium at a coastal site in Hong Kong, *Atmos. Environ.,* 33, 843-853, 1999a.

(8) Zhuang, H., Chan, C. K., Fang, M., and Wexler, A. S.: Formation of nitrate and non-sea-salt sulfate on coarse particles, *Atmos. Environ.,* 33, 4223-4233, 1999b.

***Comment 2.*** *Descriptions about the pH measurement*

*Figure S5 shows change of droplet pH during the experiment. The reviewer wonders how the measurement method of pH was validated. As described in the manuscript, pH is very important for sulfate formation. It would be beneficial to have a detailed description about pH measurement. If it were to be estimated by a thermodynamic model, it should clearly be mentioned with detailed information on how it was calculated.*

**Response:**

Since our droplets are large, we measured their pH directly based on the method reported by Craig et al. (2018). In brief, the reacted droplets were collected on pH indicator paper, and the pH was determined by colorimetric analysis of images collected immediately after sampling. We have tested the accuracy of the such method by generating pure $NH_4NO_3$ droplets equilibrated at 80% RH (Figure B). The estimated pH value by direct measurement is ~4.2, which is slightly lower than the pH value (~4.4) calculated from the E-AIM model. The error is within 5%. We have added the following text in "Method and Materials" to describe it.

**Revision in the manuscript (Section 2.2, Line 100 - 105):**

"The reacted droplets were dissolved in ~1 mL ultrapure water, followed by ion chromatograph analysis (IC analysis) with an IonPac AS15 analytical column, an AG15 guard column, and a conductivity detector. In addition, the droplets pH was measured by colorimetric analysis of images collected immediately after collection on pH indicator paper (Craig et al., 2018)."

[Figure]

**Figure B.** The direct measurement of pH of pure NH$_4$NO$_3$ droplets equilibrated at 80% RH using pH indicator paper.

"As discussed earlier, the direct reaction of NO$_2$ with S(IV) under dark is not effective in producing sulfate at 80% RH in this study. Also, no sulfate was observed during the unary uptake of SO$_2$ into NaCl droplets under dark (Figure 2a). The same was found at 60% and 70% RH (Figure 3a). Therefore, we focused on the RH dependence of sulfate production rates in the presence of irradiation. Note that 60% RH is below the deliquescence point of NaCl particles and the particles existed as supersaturated droplets."

[Figure]

**Figure C.** The morphology of NaCl droplets equilibrated at 60% RH.

*Comment 4. L136: 'However, our open system experiments with a continuous airflow (~ 0.5 L/min) would have removed the formed O3 efficiently.'*

*Do the authors have data for O3 concentration in the gas phase to support this statement?*

**Response:**

Because the number of collected droplets on the hydrophobic substrate is not large enough, it is unable to measure the $O_3$ concentration due to the technique limitation. The flow cell was used in our experiment with a dimension of 7cm (length) × 7 cm (width) × 1.5 cm (height), hence, we believe 0.5 L/min (close to 7 space volume of the reactor/min) is high enough to remove potentially formed all gas species.